# Chronic High-Fat Diet Induces Early Barrett’s Esophagus in Mice through Lipidome Remodeling

**DOI:** 10.3390/biom10050776

**Published:** 2020-05-16

**Authors:** Jeffrey Molendijk, Thi-My-Tam Nguyen, Ian Brown, Ahmed Mohamed, Yenkai Lim, Johanna Barclay, Mark P. Hodson, Thomas P. Hennessy, Lutz Krause, Mark Morrison, Michelle M. Hill

**Affiliations:** 1QIMR Berghofer Medical Research Institute, Herston, Brisbane, QLD 4006, Australia; jeff.molendijk@unimelb.edu.au (J.M.); Ahmed.Mohamed@qimrberghofer.edu.au (A.M.); 2UQ Diamantina Institute, Faculty of Medicine, The University of Queensland, Woolloongabba, Brisbane, QLD 4072, Australia; t.nguyen6@uq.edu.au (T.-M.-T.N.); y.lim2@uq.edu.au (Y.L.); M.Hodson@victorchang.edu.au (M.P.H.); thomas.hennessy@agilent.com (T.P.H.); lutz.krause@microba.com (L.K.); m.morrison1@uq.edu.au (M.M.); 3Envoi Specialist Pathologists, Herston, Brisbane, QLD 4059, Australia; IanBrown@envoi.com.au; 4Faculty of Medicine, The University of Queensland, Brisbane, QLD 4072, Australia; 5Mater Research Institute, The University of Queensland, Woolloongabba, Brisbane, QLD 4059, Australia; johanna.barclay@unsw.edu.au; 6Metabolomics Australia, Australian Institute for Bioengineering and Nanotechnology, The University of Queensland, St Lucia, Brisbane, QLD 4072, Australia; 7School of Pharmacy, The University of Queensland, Woolloongabba, Brisbane, QLD 4102, Australia; 8Agilent Technologies, Mulgrave, VIC 3170, Australia

**Keywords:** lipid, lipidomics, cardiac metaplasia, Barrett’s esophagus, esophageal adenocarcinoma, microbiota

## Abstract

Esophageal adenocarcinoma (EAC) incidence has been rapidly increasing, potentially associated with the prevalence of the risk factors gastroesophageal reflux disease (GERD), obesity, high-fat diet (HFD), and the precursor condition Barrett’s esophagus (BE). EAC development occurs over several years, with stepwise changes of the squamous esophageal epithelium, through cardiac metaplasia, to BE, and then EAC. To establish the roles of GERD and HFD in initiating BE, we developed a dietary intervention model in C57/BL6 mice using experimental HFD and GERD (0.2% deoxycholic acid, DCA, in drinking water), and then analyzed the gastroesophageal junction tissue lipidome and microbiome to reveal potential mechanisms. Chronic (9 months) HFD alone induced esophageal inflammation and metaplasia, the first steps in BE/EAC pathogenesis. While 0.2% deoxycholic acid (DCA) alone had no effect on esophageal morphology, it synergized with HFD to increase inflammation severity and metaplasia length, potentially via increased microbiome diversity. Furthermore, we identify a tissue lipid signature for inflammation and metaplasia, which is characterized by elevated very-long-chain ceramides and reduced lysophospholipids. In summary, we report a non-transgenic mouse model, and a tissue lipid signature for early BE. Validation of the lipid signature in human patient cohorts could pave the way for specific dietary strategies to reduce the risk of BE in high-risk individuals.

## 1. Introduction

There are two main forms of esophageal cancer: esophageal squamous cell carcinoma and esophageal adenocarcinoma (EAC) [1]. Over a period of three decades, the incidence of EAC has risen sixfold, while esophageal squamous cell carcinoma has remained relatively stable [2,3]. In the United States, incidence of EAC was estimated to increase from 0.40 to 2.58 cases per 100,000 between 1975 and 2009 [4]. From less than 5% of all esophageal cancer cases before the mid-1970s [5], EAC now represents almost half of all cases [2,3], making it one of the most rapidly increasing cancers in Western populations. Despite recent advances in surveillance and treatment protocols, the prognosis for patients with advanced EAC is poor, with a 5 year survival rate of less than 16%, and a median survival of less than 1 year [6,7].

EAC is widely accepted to develop via a stepwise sequence, as a consequence of gastroesophageal reflux disease (GERD). GERD leads to chronic inflammation in the esophagus and reflux esophagitis [8]. In ~10%–15% of GERD patients, the damaged squamous epithelium of the distal esophagus is replaced by cardiac mucosa with intestinal metaplasia, a condition termed Barrett’s esophagus (BE) [9,10]. Although BE itself has limited adverse health effects, patients with BE have a 30–60-fold increased risk of developing EAC [11], with estimated annual progression rate of ~0.1%–0.5% per year [12,13].

In addition to GERD and BE, epidemiology studies have identified male gender, tobacco smoking and obesity as risk factors for EAC [14]. To investigate the causality and to delineate the molecular mechanisms of GERD, surgical rodent models have been reported [15], but with high mortality rates due to the challenging surgeries. An alternative approach using dietary intervention was reported by Quante et al. [16], using 0.2% deoxycholic acid (DCA) in drinking water as a mimic of GERD to induce Barrett’s-like metaplasia in interleukin 1β transgenic mice. A follow-up study showed that high-fat diet (HFD) accelerated tumor development in the interleukin 1β transgenic mouse model [17]. While the authors report an increased inflammatory tumor microenvironment and altered intestinal microbiome as potential mechanisms, HFD may also promote EAC through lipid dyshomeostasis and esophageal dysbiosis. Circumstantial evidence suggests roles for both lipids and the esophageal microbiome in BE/EAC pathogenesis. Patients receiving cholesterol-lowering statin therapy exhibit reduced incidence of BE [18,19] and EAC [20,21,22,23]. Alterations to the esophageal microbiome have been reported in human esophageal tissues during BE/EAC disease progression [24,25], while gastric *Helicobacter pylori* infection, or altered gastric microbiota, may influence EAC development by modulating refluxate composition or frequency [26,27].

To evaluate the impact of obesity and/or GERD on esophageal tissue morphology, and to address the hypotheses that the pathogenic mechanisms of HFD or GERD involve esophageal microbiome and/or tissue lipids, we employed HFD dietary intervention and 0.2% DCA exposure in non-transgenic mice, to mimic obesity and GERD, respectively. The mouse model mimicking early BE was adapted from a previous report using BE transgenic mouse [16,17]. We found that a 9 month HFD increased esophageal tissue inflammation and cardiac metaplasia. DCA in drinking water increased the severity of HFD-induced esophageal inflammation and metaplasia segment length, potentially via increased esophageal microbiome diversity. Tissue lipidomics analyses revealed a phospholipid and sphingolipid signature associated with esophageal inflammation and cardia development.

## 2. Materials and Methods

### 2.1. Animal Experiments

The study was approved by The University of Queensland Animal Ethics Committee.

#### 2.1.1. Materials

Chow diet (Irradiated Rat and Mouse Diet) and HFD (SF04-001) were obtained from Specialty Feeds (Western Australia). Both diets were produced as cylindrical pellets with a diameter of 12 mm and comparable fiber contents of 5.2% and 5.4% respectively. The standard chow provides 12% of digestible energy from fat, 23% from protein and 65% from carbohydrates, and contained 0.78% saturated fats, 2.06% monounsaturated fats and 1.88% polyunsaturated fats by weight. The HFD provides 43% of calories from fat, 21% from protein and 36% from carbohydrates, and contained 10.03% saturated fats, 8.24% monounsaturated fats and 5.11% polyunsaturated fats by weight. Both diets were wheat- and soy-based, but differed in the primary source of fat; namely, fish meal, mixed vegetable oils and canola oil for the standard chow, or lard and soybean oil for the HFD. Deoxycholate was obtained from Sigma (Missouri, USA).

#### 2.1.2. Dietary Treatments

Eight-week-old male C57BL/6 mice were randomly assigned to one of four treatment groups for 9 months (*n* = 12).

ad libitum standard chow diet and drinking waterad libitum standard chow diet and 0.2% deoxycholic acid (unconjugated bile acid, pH 7) in drinking water [16]ad libitum HFD and drinking waterad libitum HFD and 0.2% deoxycholic acid in drinking water.

Mice were housed in groups in autoclaved standard shoe-box cages in a ventilated rack system. Drinking water with or without deoxycholate was prepared and replaced fresh weekly. All interventions were performed during the light period of a 12 h/12 h light/dark cycle.

#### 2.1.3. Tissue and Serum Collection

Tissue was collected within the same 3 h window to avoid discrepancies due to circadian variations. Blood was collected via cardiac puncture under isoflurane anesthesia followed by cervical dislocation. Blood was centrifuged at 5000× *g* for 10 min at 4 °C, and serum removed and stored at –80 °C. Distal esophagus and gastroesophageal junction tissues were collected from each mouse. The entire gastroesophageal junction was fixed for histology, while distal esophageal tissues were cut in half lengthwise. One half was fixed in formalin for histology, and one half snap frozen in liquid nitrogen for 16S ribosomal DNA (rDNA) sequencing for microbiome analysis.

#### 2.1.4. Histology

Tissues were fixed in 10% formalin for 24 h and embedded in paraffin. Embedded tissue blocks were cut into 4 µm sections and used for hematoxylin and eosin (H&E) staining. Histological evaluation and grading was performed by a specialist gastrointestinal pathologist (IB). For grading, inflammation was graded on a scale of 0 to 3 (0 = nil inflammation; 1 = mild; 2 = moderate; and 3 = severe). The presence and length of cardiac-type mucosa was recorded.

### 2.2. Lipidomics Experiments

#### 2.2.1. Materials

SPLASH LipidoMix Mass Spec Standard mixture (#330707), containing deuterated lipids of 14 species at various concentrations, and the Ceramide/Sphingoid Internal Standard Mixture II (#LM-6005), were purchased from Avanti Polar Lipids, Inc. (Alabaster, U.S.A). ESI-L low concentration tuning mix (#G1969-85000) was purchased from Agilent Technologies (Mulgrave, VIC, Australia).

#### 2.2.2. Lipid Extraction

All steps except for sonication and sample blowdown were performed on ice. Serum and tissue samples were homogenized differently but lipids were extracted using the same methyl-tert-butyl ether (MTBE)/methanol extraction method [28]. 

Mouse serum (30 µL) was added to 215 µL of ice-cold methanol containing 50 µg/mL butylated hydroxytoluene (BHT). Samples were homogenized by three rounds of vortex mixing for 30 s, freezing in liquid nitrogen for 1 min, thawing for 2 min and sonicating for 10 min at 15 °C, power 100% in a Grant XUB18 bath sonicator.

Tissue wet weight was determined using a Mettler-Toledo XS105 balance (Mettler-Toledo, Melbourne, Australia). Biopsies were transferred to Eppendorf tubes containing 500 µL ice-cold methanol, 50 µg/mL BHT and one steel bead and homogenized in a TissueLyzer LT (Qiagen, Melbourne, Australia) for six minutes at 50 Hz. Homogenate was transferred to new tubes and the original tube was washed with 400 µL methanol and transferred. Samples were dried down under nitrogen flow and resuspended in 20 µL water and 200 µL methanol (50 µg/mL BHT). Samples were homogenized by three rounds of vortex mixing for 30 s, freezing in liquid nitrogen for 1 min, thawing for 2 min and sonicating for 10 min at 15 °C, power 100% in a Grant XUB18 bath sonicator.

SPLASH LipidoMix Mass Spec Standard (10 µL) and Cer/Sph mixture II (10 µL) internal standards mixes from Avanti Polar Lipids were then added to each sample. After overnight incubation at −30 °C, 750 µL MTBE was added and each tube was vortex mixed for 10 s and shaken for 10 min on a tube rotator (4 °C). MilliQ water (188 µL) was then added, and the tube was vortex mixed for 30 s to form a biphasic separation. After centrifuging for 15 min at 15,000× *g*, 700 µL of the clear upper phase containing lipids in MTBE was transferred to another tube and dried down using a gentle stream of nitrogen. After drying down of lipids, extracts were resuspended in 50 µL methanol (containing 50 µg/mL BHT)/toluene (90%/10%, *v*/*v*). Dry weight of the remaining pellets from tissue samples was determined in triplicate using a Mettler-Toledo XS105 balance. Dry weights were used to normalize lipid injection volumes of tissue samples prior to mass spectrometry analysis. For serum samples equal volumes were injected.

#### 2.2.3. Untargeted Lipidomics

An Agilent Technologies 1290 Infinity II UHPLC system with an Agilent ZORBAX Eclipse plus C18 1.8-micron column (#959757-902) and guard column (#821725-901), coupled online to an Agilent 6550A iFunnel QTOF mass spectrometry system, was used for untargeted lipidomics. The mass spectrometer was tuned in the low mass range (1700 *m/z*), high sensitivity slicer mode and the instrument mode was set to Extended Dynamic Range (2 GHz). The quadrupole and time-of-flight (TOF) sections of the mass spectrometer were both tuned prior to each experiment. The quadrupole was tuned to reference masses 118.09, 622.03 and 1221.99 in positive ionization mode. Experiments were performed if the quadrupole component passed the check tune for each reference mass in wide, medium and narrow modes. The TOF component was tuned using reference masses 118.09, 322.05, 622.03, 922.00, 1221.99 and 1521.97 in positive ionization mode. TOF mass calibration indicated that at around 110–120 *m*/*z* the resolution was ~12,000–13,000 and increased to 20,000–21,000 around 600–620 *m/z* range. The ion source used was Dual Agilent Jet Stream electrospray ionization, which allows for the simultaneous introduction of sample and reference masses into the mass spectrometer. Source capillary voltages were set to 4000 V for positive ionization mode whilst the nozzle voltage was set to 0 V, fragmentor was set to 365 and octopoleRFPeak to 750. Nitrogen gas temperature was set to 250 °C at a flow of 15 L/minute and a sheath gas temperature of 400 °C at a flow of 12 L/min. During the experiment reference masses were enabled (121.05 and 922.01 Da) to enable auto-recalibration of compounds with known masses. MS1 data was acquired between 100–1700 *m/z* at a scan rate of 2.5 spectra per second.

The sample dilution and injection volume used for experiments was determined by testing a representative sample prior to analyzing the cohort. Reversed phase buffers A and B contained 25 millimolar (mM) ammonium formate and 0.1% formic acid in 60%/40% (*v*/*v*) acetonitrile/water or 90%/10% (*v*/*v*) isopropanol/water respectively. The separation gradient was run at a flow rate of 0.5 mL/min to separate the lipids during a 16 min gradient. The method started at 15% B and increased to 30% B at 2:00, 48% B at 2:30, 82% B at 11:00, 99% B at 11:30. The gradient was retained at 99% B until 13:00 and retained at the starting condition of 15% B between 13:06 and 16:00. The column compartment was maintained at 60 °C for the duration of the experiment.

#### 2.2.4. Targeted Lipidomics

Targeted lipidomics were performed on an Agilent Technologies 1290 Infinity UHPLC system with an Agilent HILIC Plus RRHD 2.1 × 100 mm 1.8 micron column, coupled online to an Agilent 6490A Triple Quadrupole mass spectrometer with iFunnel and Agilent Jet Stream electrospray ionization source, operated in dynamic MRM mode. The source nitrogen gas temperature was set to 250 °C at a flow rate of 15 L/min, and the sheath gas temperature set to 400 °C at a flow rate of 12 L/min. The capillary voltage was set to 4000 V for positive mode and 5000 V for negative mode and the nebulizer operated at 30 psi. Ion funnel low and high pressure in positive mode were 150 and 60, and in negative mode 150 and 120, respectively. Check tunes were performed in wide, unit and enhanced modes prior to each experiment to confirm the performance of the mass spectrometer. The quadrupole was tuned to reference masses 118.09, 322.05, 622.03, 922.01 and 1221.99 in positive ionization mode, and 112.99, 302.00, 601.98, 1033.99 and 1333.97 in negative ionization mode.

Each sample was analyzed in 3 separate dynamic MRM runs using two different HILIC buffer systems, both using 50%/50% (*v/v*) acetonitrile/water as Buffer A and 95% acetonitrile/water (*v/v*) as buffer B. The buffers were supplemented with 25 mM ammonium formate, pH 4.6 and 0.1% formic acid (denoted methods F1, F2) or 10 mM ammonium acetate, pH 7.6 (denoted method A). As detailed in Appendix A, the methods had 155 (F1), 156 (F2) and 126 (A) transitions, including internal standards. The minimum dwell times were 4.2 milliseconds (ms), 4.1 ms and 3.1 ms respectively for methods F1, F2 and A. The method started at 0.1% A and increased to 40% A at 8:00, 90% A at 9:30 until 10:30. The gradient decreased to 0.1% A between 10:30 and 11:30 and was retained at the starting conditions of 0.1% A until 14:00. The column compartment was maintained at 30 °C for the duration of the experiment. A pooled quality control (QC) sample was injected multiple times to condition the HPLC column prior to analyzing samples, and also queued after every 6–7 biological samples to monitor mass spectrometry performance for the duration of the experiment [29,30].

#### 2.2.5. Data Treatment and Analysis

Feature integration of untargeted lipidomics data was performed using the XCMS Centwave method and retention time alignment was performed using the Obiwarp method [31]. Features were grouped and peak filling was performed using the fill ChromPeaks method. Finally, feature information and abundances per samples were exported as a .csv file format. Lipid identification was performed using MS-DIAL version 3.90 (RIKEN Center for Sustainable Resource Science, Kanagawa, Japan) and the included FiehnRT (v47) lipid database [32]. Identifications were made based on accurate mass, retention time and database matching, and then manually confirmed. The MS1 tolerance was set to 0.01 Da and the tolerance for MS2 peaks was set to 0.05 Da. Database retention times were not used for scoring in the lipid identification. An identification score cut-off of 70 was set to remove most inaccurate identifications. The possible adduct ions were set to [M + H]^+^, [M + NH_4_]^+^ and [M-H]^−^. Manual confirmation included the visual inspection of all database matches, assessing the dot and reverse dot product similarity scores. Ambiguous identifications of features with multiple likely identifications were excluded from the analysis. Lipid identifications, accurate masses and retention times were exported from MS-DIAL and integrated into the data exported from XCMS.

Acquired targeted lipidomics data was imported into Skyline (MacCoss Lab, Department of Genome Sciences, University of Washington) [33], peak integration was automated but manually confirmed and corrected if required. Internal standard retention time was used to confirm correct peak integration of lipids belonging to the same class. Peak areas were exported from Skyline for further analysis in R (R Foundation for Statistical Computing, Vienna, Austria) [34].

The datasets were filtered to remove any lipids with a coefficient of variation greater than 20% among the quality control samples. Missing values were imputed using the MinDet method from the imputeLCMD R package using the default q-value of 0.01. All datasets were log2 transformed and normalized using the probabilistic quotient normalization method as described by Dieterle et al. [35]. Lipid information such as lipid class, number of unsaturated bonds and fatty acid chain lengths were parsed from the original lipid names using the lipidr R package [36]. Further analyses and visualizations, including principal component analysis (PCA) and lipid class boxplots were produced using lipidr [36]. The enrichment of lipid classes was determined using the LSEA (lipid set enrichment analysis method) [36]. Pearson correlation was used to determine the correlation between total lipid fatty acid chain lengths and the development of disease conditions.

### 2.3. Microbiome Profiling

#### 2.3.1. DNA Extraction

Unless otherwise stated, solvents were purchased from Sigma (Missouri, USA). Mouse tissues were preincubated with lysis buffer (20 nanomolar (nM) Tris/HCl; 2 mM EDTA; 1% Triton X-100; pH 8; supplemented with 20 mg/mL lysozyme) for 60 min at 37 °C, then with 25 µL Proteinase K (20 mg/mL; Ambion, CA, USA) at 56 °C until completely lysed. DNA was extracted using the ISOLATE II Genomic DNA Kit (Bioline, London, UK) following manufacturer’s standard protocol. The DNA samples were eluted in two lots of 50 µL Elution Buffer G from the kit.

#### 2.3.2. Library Preparation and Sequencing

Library preparation was performed in batch. Polymerase chain reaction (PCR) preparation was conducted in a designated DNA template-free room. Sequencing library preparation of the samples and control (no DNA template) was based on the 16S Metagenomic Sequencing Library Preparation guidelines provided by Illumina. Q5 Hot Start High-Fidelity 2× Mastermix polymerase (NEB, Ipswich, MA, USA) was used for the Amplicon PCR step. Primers used for the amplification of the V6–V8 region of the 16S ribosomal RNA gene were primers 927-Forward (AAACTYAAAKGAATTGRCGG; universal) and 1392-Reverse (ACGGGCGGTG WGTRC; universal) with Illumina adapter sequences. Samples were barcoded using the Illumina dual-index system (Nextera XT v2 Index Kit Set A) for the Index PCR step. PCR products were purified using AMPure XP beads (Beckman Coulter, Brea, CA, USA). The DNA concentration for each barcoded amplicon mixture was quantified following manufacturer’s instructions (Quantus, Promega, Madison, WI, USA) and all samples were pooled to provide 4 nanomol of each amplicon. The pooled libraries were sequenced using the Illumina MiSeq platform (Illumina, San Diego, CA, USA) and the MiSeq Reagent Kit v3 (2 × 300 bp) by the Australian Centre for Ecogenomics, located at the University of Queensland.

#### 2.3.3. Bioinformatics and Statistical Analysis

Raw sequencing reads were processed and analyzed using Quantitative Insights Into Microbial Ecology 2 (QIIME 2, version 2019.7) according to the developer’s recommendations [37]. Sequence quality control was carried out using the DADA2 algorithm, a QIIME 2 plugin-software to filter low-quality sequences as well as to identify and remove chimeric sequences. Amplicon sequence variants (ASVs) were generated from the filtered sequences and the SILVA_132 99% reference database was used to train the feature classifiers and provide taxonomic assignment accordingly. An ASV table was generated and normalized using total sum normalization (TSS) for all further analyses using Calypso (version 8.84) [38].

## 3. Results

### 3.1. High-Fat Diet and Bile Acid Exposure as a Mouse Model for the Development of Esophageal Inflammation and Cardiac Metaplasia

Chronic treatment with the unconjugated bile acid, deoxycholic acid (DCA, 0.2%), in drinking water was previously reported to accelerate Barrett’s-like metaplasia development in an interleukin-1β transgenic mouse model [16]. We hypothesized that obesity induced by chronic HFD will replicate the chronic inflammation due to interleukin-1β overexpression, and leads to Barrett’s-like epithelium development in wild-type mice. To test this hypothesis, male C57BL/6 mice were fed with standard chow diet or HFD with and without 0.2% DCA, for 9 months prior to sacrifice (*n* = 11 per group). Chow and HFD diets had comparable fiber (5.2% vs 5.4%) and protein (23% vs 21%) content, but the digestible energy from fat increased from 12% in chow to 43% in HFD, while carbohydrate reduced from 65% to 36%.

Body weight was monitored weekly, and HFD +/− DCA mice had significantly higher body weight than Chow +/− DCA (*q* < 0.0001), but no difference in body weight was observed between mice +/− DCA in either diet group (Figure 1a). Interestingly, weight gain in mice in the HFD + DCA group was delayed compared to the HFD + water group, potentially indicative of DCA-induced esophageal damage reducing food intake and subsequent recovery (Figure 1a).

Next, we examined the impact of the dietary treatments on tissue morphology of the gastroesophageal junction, where BE arises. H&E stained tissues were evaluated, and graded for inflammation severity and metaplasia length by an expert gastrointestinal pathologist in a blinded manner. Figure 1b shows the morphology of the normal squamous epithelium of the gastroesophageal junction, which was observed in most samples. In contrast, cardiac metaplasia with neutral mucin-producing glands was observed immediately adjacent to the squamous epithelium, observed in all four groups with varying frequency. Furthermore, varying grades of inflamed esophageal tissue were observed (Figure 1c). Inflammation grade 0 lacks inflammatory cells in the lamina propria, whereas mild inflammation with small numbers of lymphocytes and eosinophils are observed in inflammation grade 1. Inflammation grade 2 is marked by moderate inflammation, with a prominent infiltration of the lamina propria by lymphocytes and small numbers of eosinophils. Additionally, lymphocytes infiltrate the squamous epithelium. In severe inflammation, grade 3, a prominent infiltration of the lamina propria by lymphocytes, plasma cells, eosinophils and neutrophils is observed. Neutrophils and eosinophils are present within the epithelium.

Quantitative analysis revealed a basal level of mild inflammation in ~20% of the control and DCA treatment groups (Figure 2a). The combined HFD + DCA increased the overall incidence of inflammation to 67%, and was the only group with a grade of severe inflammation (Figure 2a). HFD alone slightly increased inflammation incidence to 27%, but induced a similarly high level of metaplasia (64%–67%) as the combined HFD + DCA (Figure 2b). However, all of the instances of metaplasia for the HFD + DCA group were long segment, while metaplasia induced by HFD alone comprised short, medium and long segments (Appendix A).

The above results demonstrate that chronic HFD with DCA (mimicking GERD) induces the hallmarks of early BE, namely, tissue inflammation and metaplasia. To further evaluate the correlation between each dietary treatment, we next asked if the inflammation or metaplasia incidence correlate with HFD or DCA treatment. When all samples from DCA treatment groups were compared against all groups treated with water, no significant difference was detected for incidence of cardiac metaplasia (Figure 2c). Similarly, HFD, with or without DCA, did not significantly increase the development of cardiac metaplasia (Figure 2d). Finally, we asked whether the incidence of inflammation and metaplasia was correlated, and found a significant relationship, with 6% of mice without esophageal inflammation and 54% of mice with inflammation developing cardiac metaplasia (Figure 2e). Furthermore, among the mice that developed cardiac metaplasia, the mice with inflammation developed a longer metaplastic tissue (Figure 2e).

### 3.2. Esophageal Tissue Microbiome Diversity Increases with DCA

After confirming the induction of gastroesophageal junction inflammation and cardiac metaplasia by chronic HFD + DCA treatment, we went on to profile the esophageal microbiota of 43 samples from the four study groups, using 16S ribosomal RNA gene sequencing. One sample gave no sequences and was removed from subsequent analysis. In total, 21,708 high quality sequences were obtained, with an average of 504.84 sequences per sample. From these sequences, four major phyla (*Actinobacteria*, *Bacteroidetes*, *Firmicutes*, and *Proteobacteria*) were identified, and a total of 23 ASVs were detected at 99% sequence identity threshold via SILVA_132 database.

We first compared microbial diversity (Shannon index) and richness between treatment groups using rank test in Calypso. No significant differences in microbial richness was observed, but a higher microbial diversity was observed in DCA alone, and HFD + DCA groups (Figure 3a). To further test the relationship between DCA and microbial diversity, we then re-grouped the data into HFD-treated and DCA-treated groups, as previously done (Figure 2). While no significant differences in microbial diversity or richness were detected for HFD treatment (Figure 3b), a significant increase of microbial diversity in DCA-treated groups was detected, with a similar but non-significant increase in richness (Figure 3c).

### 3.3. Lipidomic Changes Associated with Dietary Interventions

In parallel to the esophageal microbiome analysis, we conducted lipidomics analyses on the collected serum and gastroesophageal junction samples, to determine associations between the respective lipidomes and dietary treatments (HFD or DCA), inflammation or cardiac metaplasia. A combined approach of untargeted and targeted lipidomics was conducted, to quantitate 339 and 197 mammalian lipid species in the serum and gastroesophageal junction samples, respectively.

While we observed no separation of gastroesophageal junction lipidome as a result of dietary treatments by PCA in the first two principal components (Figure 4b), the serum lipidome showed clear separation and clustering according to dietary intervention groups (Figure 4a).

Differential expression analysis was conducted on the lipidomics data of both datasets. Lipid class enrichment was conducted to determine if specific lipid classes were selectively altered. The boxplots in Figure 4c,d summarize the log_2_ fold change for each lipid class for each group, for serum and gastroesophageal junction tissue lipids, respectively. Statistically significant changes are colored in blue. Gastroesophageal junction tissue lipid class analysis (Figure 4d) revealed overlapping impacts of HFD and DCA treatments. All three treatment groups showed elevated lysophosphatidylcholine (LPC), as well as decreased phosphatidylcholine (PC) and phosphatidylethanolamine (PE) (Figure 4d). While triacylglycerol (TAG) was elevated only in group B (DCA alone), phosphatidylglycerol (PG) was elevated in HFD-treated groups (Figure 4d). For serum lipids, both HFD-treated groups (C and D) show similar changes, with elevated ceramide (Cer), PG and sphingomyelin (SM), and reduced lysophosphatidylethanolamine (LPE), PE and phosphatidylinositol (PI) (Figure 4c). In contrast, DCA treatment alone (Group B) showed a large decrease in ether-PC, with modest changes in PI and SM (Figure 4c). Interestingly, the reduction in ether-PC was not observed in the combined HFD + DCA treatment (Group D), suggesting HFD rather than DCA is the main driver of the serum lipidome.

### 3.4. Lipidomic Changes Associated with Early Tissue Pathology

Since esophageal inflammation or metaplasia occurred in ~10% to 70% of mice in each group, we next investigated the association between serum and gastroesophageal junction tissue lipidome with early esophageal pathology. To this end, lipid class enrichment analysis was conducted on metaplasia vs normal samples, and inflamed vs normal samples. Apart from elevated serum ether-PC, the serum lipidome returned minor changes of < 25% magnitude (Figure 5a). In contrast, the tissue lipidome showed similar changes for metaplasia and inflammation, characterized by reduced lysolipids and elevated ceramides (Figure 5b). This result revealed major differences between the lipidome associated with dietary intervention (Figure 4) and that associated with esophageal pathology (Figure 5). Specifically, differences were observed for ceramides and the lysolipids LPC and LPE. Elevated tissue ceramide was associated with metaplasia and inflamed tissue, but not with any dietary treatment, even in the HFD + DCA treatment group, where 66.7% of cases were inflamed (Figure 5). Reductions in the lysolipids LPC and LPE were associated with metaplasia and inflammation (Figure 5b), but elevated tissue LPC was associated with HFD and DCA treatment (Figure 4d). These results strongly implicate roles for elevated ceramides and reduced lysolipids in metaplasia development due to chronic inflammation.

As differing fatty acid chain lengths on a lipid can greatly impact biological function in cancer development [39], we next determined whether fatty acid chain lengths were associated with inflammation or metaplasia for the Cer, LPC and LPE classes. Ceramides comprise a single fatty acid chain with a sphingoid backbone (commonly 18:1, as illustrated in Figure 6a). Figure 6a plots the log2 fold change for different total fatty acid chain lengths of each measured ceramide species. As evident in Figure 6a, a significant correlation was found between very long chain ceramides and the disease conditions inflammation and metaplasia. On the other hand, specificity in fatty acid chain lengths were not observed for LPC in either metaplasia or inflamed tissues (Figure 6b). Increased LPE chain lengths were significantly correlated with metaplasia, but not with inflammation (Figure 6c).

## 4. Discussion

This is the first study to demonstrate that chronic HFD in non-transgenic mice is sufficient to induce esophageal inflammation and cardiac metaplasia, the first steps in BE/EAC pathogenesis. While DCA in drinking water had no effect on esophageal morphology on its own, it increased the severity of inflammation and length of metaplasia when combined with HFD. HFD clearly induced obesity and serum lipid derangements, but only a proportion of HFD-treated mice developed esophageal inflammation and cardiac metaplasia. Intriguingly, the esophageal tissue lipidome showed a similar signature for inflammation and metaplasia, which was not associated with HFD. These results suggest that homeostatic mechanisms can buffer HFD/obesity-induced lipidome derangement to an extent, beyond which inflammation and metaplasia ensue.

Obesity increases the risk of several cancer types, and the mechanisms of specific lipids on carcinogenesis are beginning to be revealed [39]. In this study, we identified an esophageal tissue lipid signature for inflammation and metaplasia, which is characterized by elevated very long chain ceramides and reduced lysolipids, LPC and LPE. Very long chain ceramides have been reported to increase cancer proliferation, and evade growth suppressor and apoptotic signals [39]. A link between HFD and tissue ceramide levels was recently reported by Zalewska et al. [40] for submandibular gland ceramide following HFD treatment in mice. The authors suggested that elevated ceramide increased mitochondrial reactive oxygen species (ROS) production and respiratory chain, leading to inflammation [40].

Phospholipid remodeling has recently emerged as playing an important role in disease pathogenesis, through the characterization of the lysophosphatidylcholine acyltransferase (LPCAT) family [41]. Lysolipids LPC and LPE contain a single fatty acyl chain, while the more abundant PC and PE contain two fatty acyl chains. Due to the differing biophysical properties, altered lysolipid:phospholipid ratio can lead to altered membrane curvature and fluidity, which could translate to organelle remodeling and altered signal transduction in pathology [41].

Warnecke-Eberz et al. [42] identified the LPCAT1 gene to be elevated in late- and early-stage esophageal adenocarcinoma tissue, compared to adjacent normal tissue. Elevated LPCAT1 could explain the decreased LPC and increased PC that we identified for inflamed and cardia gastroesophageal junction tissue (Figure 5). LPCAT1 enzyme and LPC are elevated in several other cancers, including colorectal cancer [43], hepatocellular carcinoma [44], gastric cancer [45] and clear cell renal carcinoma [46]. Interestingly, body fatness is a risk factor for all these cancers [47]. In a recent study of western diet-associated non-alcoholic steatohepatitis, LPCAT1 and LPCAT2 are in the top 10 liver genes/transcripts most significantly elevated in mice fed western style diets compared to standard diets [48]. Together, these data suggest a mechanistic link between high-fat diet, activation of LPCAT transcripts, altered LPC:PC ratio, and induction of esophageal inflammation and metaplasia development.

As GERD is a well-established risk factor for BE, the lack of esophageal pathology from the mice treated with DCA alone was somewhat surprising. This result may suggest that 0.2% DCA in drinking water does not fully mimic GERD, or that GERD is less damaging to mice esophagus compared to human. Nevertheless, as expected for the additive effect of risk factors, DCA treatment in addition to HFD increased the severity of inflammation and length of metaplasia, compared to HFD treatment alone. DCA treatment increased the esophageal microbiome diversity, which is consistent with previous reports describing the effect that levels of bile acids in the gut have on the major division/phyla level taxa of the gut microbiome [49]. These effects could potentially extend to the esophagus, given that the composition of the esophageal microbiome depends on the oral and gut microbiome [50]. Previous studies have reported a depletion of Gram-positive *Streptococcus*, and enrichment of Gram-negative taxa, including *Veillonella* and *Prevotella*, in BE [25,51]. Interestingly, dysplasia and esophageal adenocarcinoma were reported to have reduced esophageal microbiome diversity [24]. Further studies will be required to establish the cause–effect relationship and mechanisms of esophageal microbiota in BE/EAC pathogenesis.

## 5. Conclusions

In conclusion, we report the results of a dietary intervention model for early BE, and a lipidomic signature for inflamed and metaplastic esophageal tissue. In non-transgenic mice, chronic HFD was sufficient to induce inflammation and cardiac metaplasia at the gastroesophageal junction. As a GERD-mimic, bile acid in drinking water in addition to HFD increased the severity of inflammation and length of metaplasia. GERD, but not HFD, increased the esophageal microbiome diversity. The causality of microbiome in BE development remains to be established.

## Figures and Tables

**Figure 1 biomolecules-10-00776-f001:**
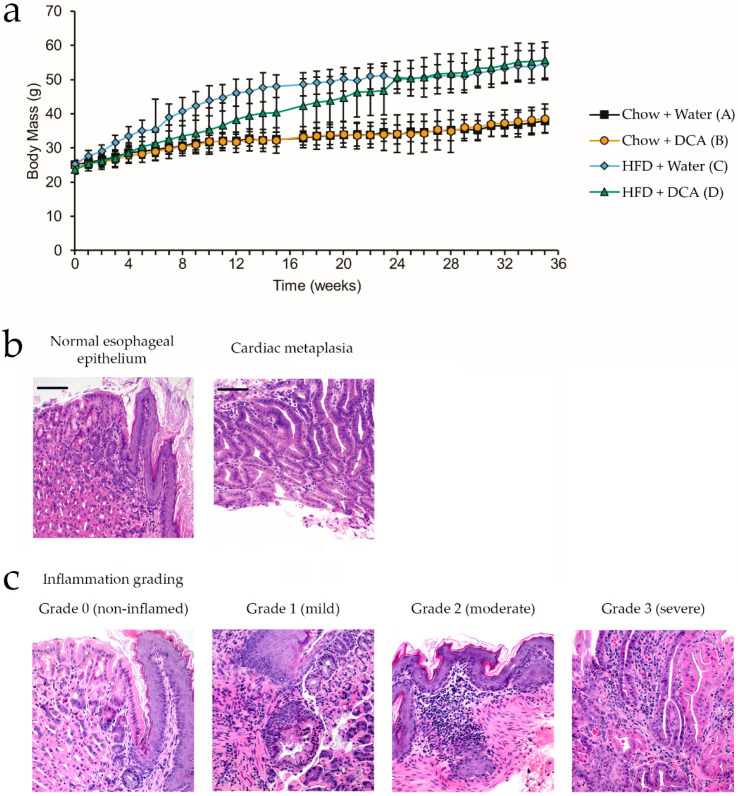
Chronic high-fat and/or bile acid dietary intervention in wild-type mice induces chronic inflammation and cardiac metaplasia development at the gastroesophageal junction. C57BL/6 mice (*n* = 11 per group) were given +/− high-fat diet (HFD) and +/− 0.2% deoxycholic acid (DCA) over a 9 month period, and gastroesophageal junction tissue morphology evaluated in hematoxylin & eosin (H&E) stained sections for inflammation and epithelial changes. (**a**) Body weight over time for each of the four groups. Values are mean ± SD; (**b**) Example esophageal epithelium morphology for normal and cardiac metaplasia. (**c**) Example inflammation grading. (200×; scale bar 100 μm).

**Figure 2 biomolecules-10-00776-f002:**
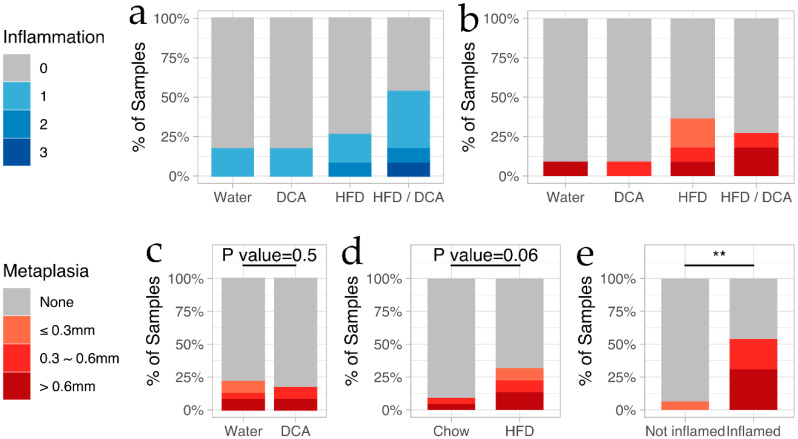
Synergistic action of chronic HFD and DCA promotes inflammation and cardiac metaplasia at the gastroesophageal junction. H&E stained tissue sections graded for the degree of inflammation (mild, moderate or severe), and the length of cardiac metaplasia (short, medium or long) were analyzed for (**a**) the occurrence and degree of inflammation, and (**b**) length of cardiac metaplasia in the four treatment groups. Correlation between presence of cardiac metaplasia was further compared for: (**c**) all mice treated with DCA compared to water control; (**d**) all mice on HFD diet compared to chow diet; and (**e**) any level of inflammation. The significance for plots c–e was calculated using the Fisher’s exact test. ** *p*-value < 0.05.

**Figure 3 biomolecules-10-00776-f003:**
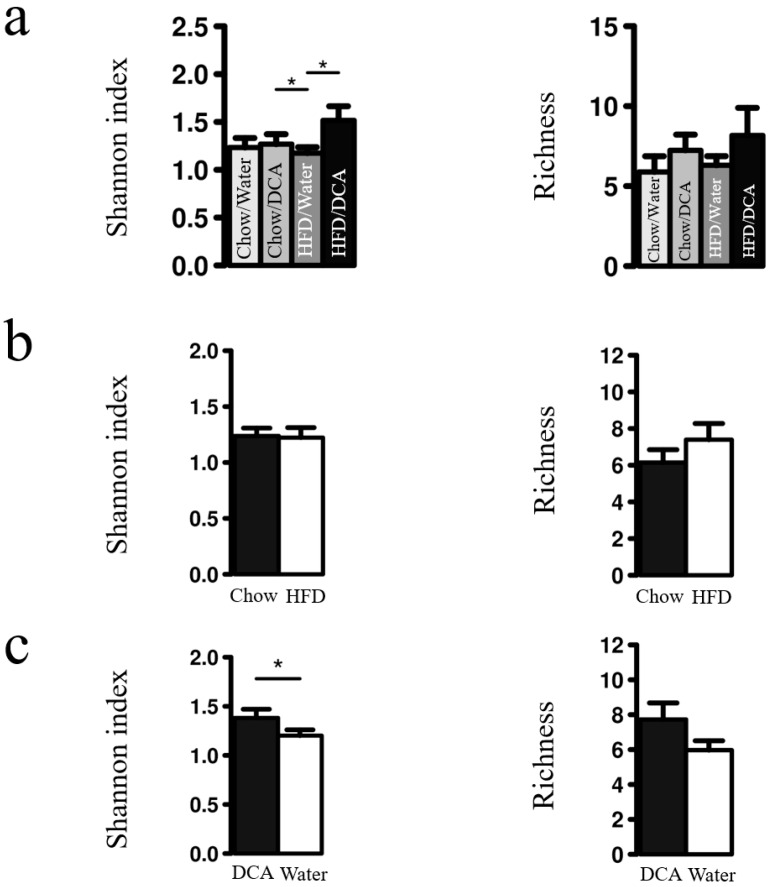
Esophageal microbiome diversity is increased by HFD + DCA treatment. Shannon index and microbial richness of esophageal microbiome data was measured using rank test for (**a**) each of the four treatment groups, (**b**) combining HFD/Water and HFD/DCA groups into the HFD group, and Chow/Water + Chow/DCA into the Chow group, or (**c**) combining Chow/DCA and HFD/DCA groups into the DCA group, and Chow/Water + Chow/DCA into the Water group. * *p*-value < 0.01.

**Figure 4 biomolecules-10-00776-f004:**
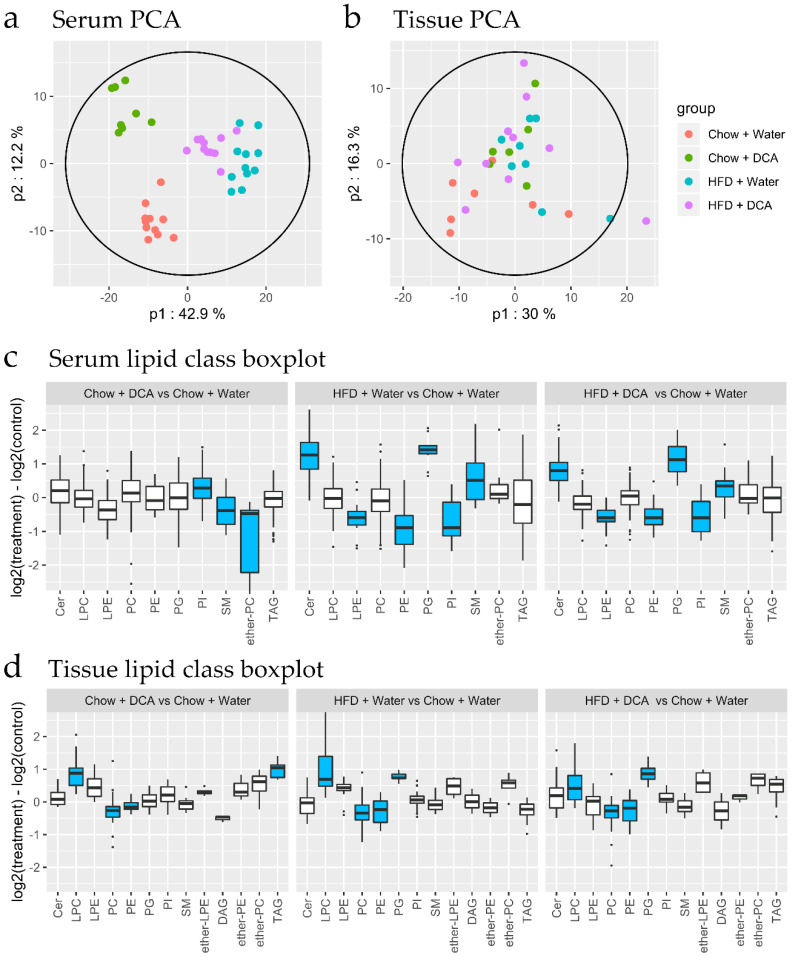
The impact of dietary interventions on tissue and serum lipidome. After 9 months high fat diet (HFD) +/- 0.2% deoxycholate (DCA), mouse gastroesophageal junction tissue and serum samples were subjected to lipidomics analyses. (**a**) Principal component analysis score plot of mean-centered unit variance-scaled untargeted serum lipidome data (*n* = 38). (**b**) Principal component analysis score plot of mean-centered unit variance-scaled untargeted gastroesophageal junction tissue lipidome data (*n* = 29). Plot ellipses represents the 95% Hotelling’s T2 confidence intervals for the multivariate data. (**c**,**d**) Lipid class boxplots for serum and gastroesophageal junction tissue lipids, showing the distribution of log2 differences between the treatment group and control. Positive values represent lipids that are more abundant in the treatment group than in the control group. Blue color indicates significant enrichment using the fast gene set enrichment analysis (fgsea) method. Cer—Ceramide, LPC—lysophosphatidylcholine, LPE—lysophosphatidylethanolamine, PC—phosphatidylcholine, PE—phosphatidylethanolamine, PG—phosphatidylglycerol, PI—phosphatidylinositol, SM—sphingomyelin, DAG—diacylglycerol, TAG—triacylglycerol.

**Figure 5 biomolecules-10-00776-f005:**
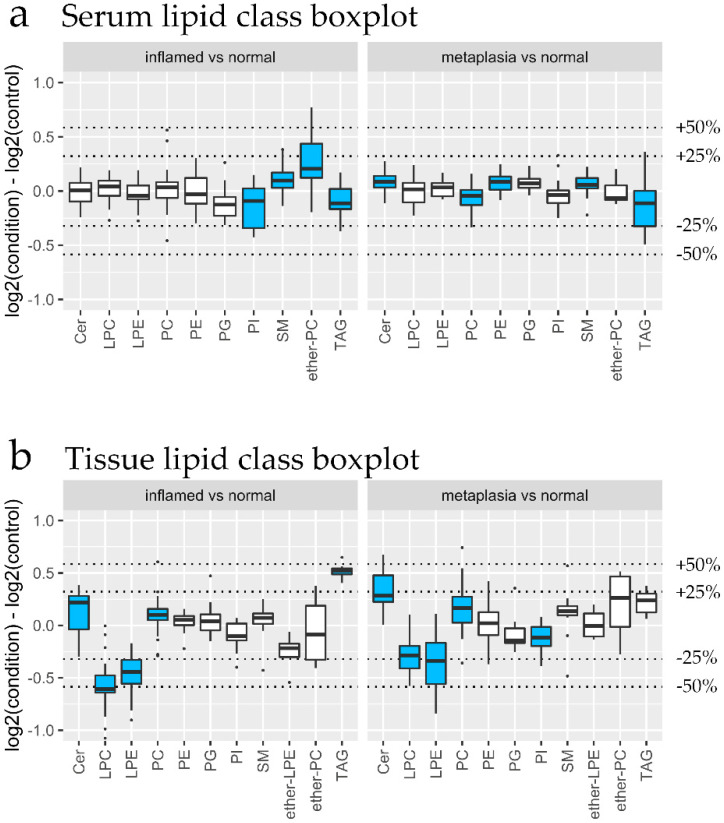
Lipid classes associated with gastroesophageal junction tissue pathology. (**a**,**b**) lipid class boxplots for serum and tissue lipids, showing the distribution of log2 differences between the disease condition and control. The disease conditions cardia and inflammation were visualized after applying the removeBatchEffect function from the limma R package. Positive values represent lipids that are more abundant in the disease condition group than in the control group. Blue color indicates significant enrichment using the fast gene set enrichment analysis (fgsea) method. Cer—Ceramide, LPC—lysophosphatidylcholine, LPE—lysophosphatidylethanolamine, PC—phosphatidylcholine, PE—phosphatidylethanolamine, PG—phosphatidylglycerol, PI—phosphatidylinositol, SM—sphingomyelin, DAG—diacylglycerol, TAG—triacylglycerol.

**Figure 6 biomolecules-10-00776-f006:**
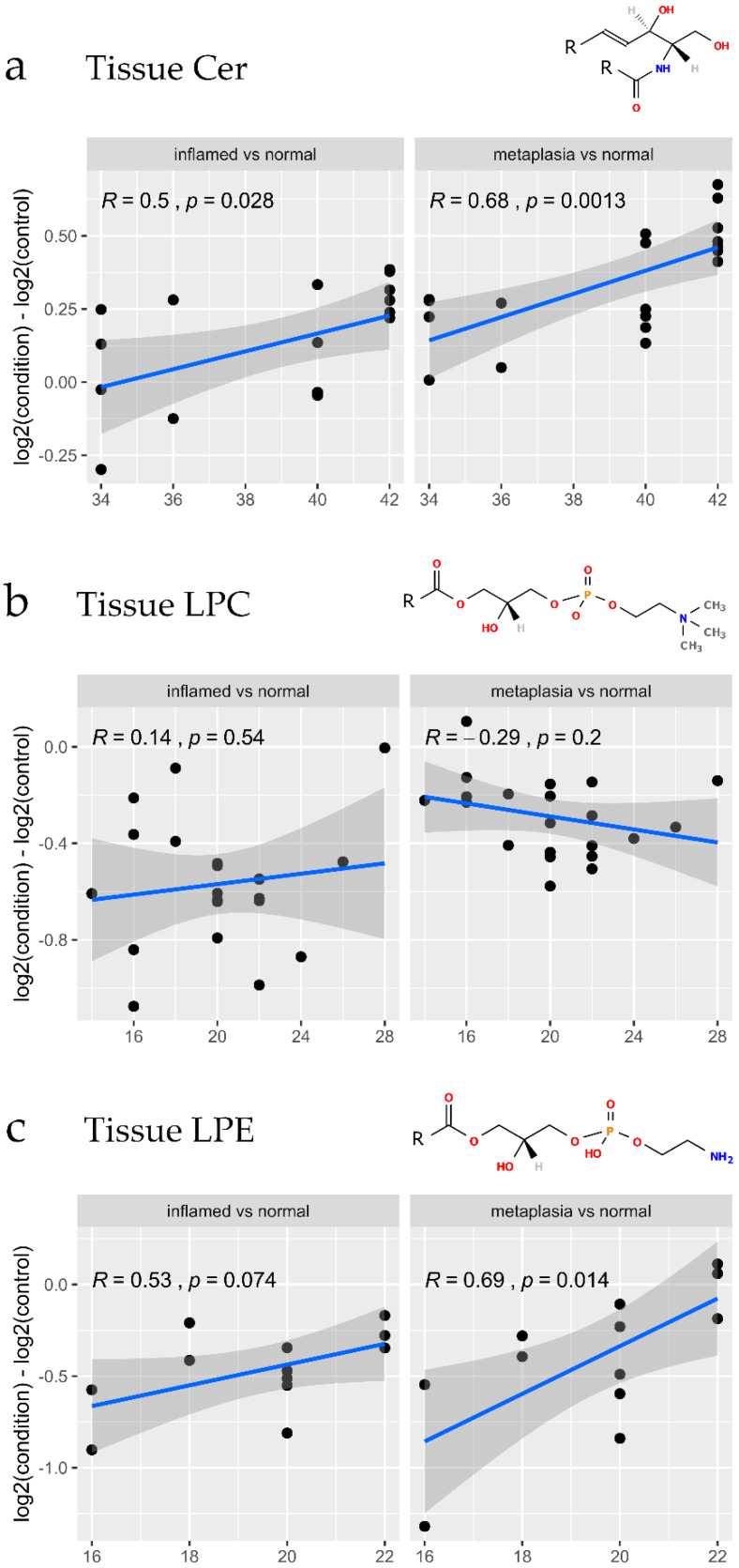
Association of tissue ceramide, LPC and LPE fatty acid chain lengths with gastroesophageal junction tissue pathology. (**a**–**c**) Total chain length plots for tissue lipids, showing the alterations in log2 abundances between the disease condition and control. The x-axis labels refer to the total fatty acid chain length of the measured lipid. The disease conditions metaplasia and inflammation were visualized after applying the removeBatchEffect function from the limma R package. Positive values represent lipids that are more abundant in the disease condition group than in the control group. The metrics shown in the plots refer to the Pearson correlation coefficient (R) and *p*-value. The smoothed line and 95% confidence interval were drawn using geom_smooth, by fitting a linear model. The structures above each plot represent lipid species of the Ceramide, LPC and LPE classes, where the R groups refer to the hydrocarbon chains of varying lengths.

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
