# Peer review of "Chronic High-Fat Diet Induces Early Barrett’s Esophagus in Mice through Lipidome Remodeling"

_biomolecules, 2020, doi:10.3390/biom10050776_

Round 1
Reviewer 1 Report
The authors demonstrate esophageal inflammation and cardiac metaplasia are induced by chronic HFD, which is a novel finding point out the link between obesity and inflammation in esophagus. They presented some very informative data potentially supporting how lipidomic works on the progression of Barret’s esophagus. Despite some interest, overall this current manuscript need to be added more evidences to support their conclusions.
- The manuscript needs to be carefully proofread for misspelling, grammar issues.
- In Figure1a, how many mice in each of the four groups? Please indicate the score of inflammation. I am confused about the figure legends from b to e, please make it more clear to readers.
- Is there any in vitro data to support authors’ conclusion?
Author Response
The authors demonstrate esophageal inflammation and cardiac metaplasia are induced by chronic HFD, which is a novel finding point out the link between obesity and inflammation in esophagus. They presented some very informative data potentially supporting how lipidomic works on the progression of Barret’s esophagus. Despite some interest, overall this current manuscript need to be added more evidences to support their conclusions.
1. The manuscript needs to be carefully proofread for misspelling, grammar issues.
Response: Thank you for the comment. We have carefully proofread the manuscript and discovered some errors that were previously missed. After correcting these errors, we are confident that the quality of the manuscript has improved.
2. In Figure1a, how many mice in each of the four groups? Please indicate the score of inflammation. I am confused about the figure legends from b to e, please make it more clear to readers.
Response: Apologies for the confusion. There were 11 mice in each group. This information is now included in the figure legend, and we also improved the labelling of the images in the figure. Furthermore, we added a detailed explanation of the inflammation scoring as well as example images for each inflammation grade (Figure 1c).
“Furthermore, varying grades of inflamed esophageal tissue were observed (Figure 1c). Inflammation grade 0 lacks inflammatory cells in the lamina propria, whereas mild inflammation with small numbers of lymphocytes and eosinophils are observed in inflammation grade 1. Inflammation grade 2 is marked by moderate inflammation with a prominent infiltration of the lamina propria by lymphocytes and small numbers of eosinophils. Additionally, lymphocytes infiltrate the squamous epithelium. In severe inflammation, grade 3, a prominent infiltration of the lamina propria by lymphocytes, plasma cells, eosinophils and neutrophils is observed. Neutrophils and eosinophils are present within the epithelium.”
3. Is there any in vitro data to support authors’ conclusion?
Response: Unfortunately, the relationship between obesity/reflux, esophageal inflammation and Barrett’s esophagus involves a complex interplay of various cell types and glands, which has not yet been mimicked in vitro [1]. Furthermore, some concerns have been raised on the phenotype of the available immortalized Barrett’s esophagus and squamous esophageal epithelial cell lines [1, 2]. Therefore, the mouse dietary intervention appeared to be the most suitable experimental model to examine the role and mechanisms of obesity and reflux in early Barrett’s esophagus development.
References:
- Ahrens, T.D., et al., Turning Skyscrapers into Town Houses: Insights into Barrett's Esophagus. Pathobiology, 2017. 84(2): p. 87-98.
- Underwood, T.J., et al., A comparison of primary oesophageal squamous epithelial cells with HET-1A in organotypic culture. Biol Cell, 2010. 102(12): p. 635-44.
Reviewer 2 Report
Please supply additional information with regard to the dietary treatments. Descriptive terms such as "high fat diet" and "standard chow" seem singularly uninformative, given the evidently fundamental importance of the differing dietary exposures in this investigation. It therefore seems essential to provide and briefly discuss some further information about the composition of the diets. For example, what are the sources of protein and carbohydrate in the two diets and how do they differ? Do the dietary treatments have similar physical characteristics? Presumably the diets differ in dietary fibre content? Could this be an important determinant of the microbiome, even in the upper gut? What is the lipid composition of the "standard chow"? Any further information that can be given about differences in fatty acid composition would be valuable.
Author Response
Thank you for raising this important point. Detailed composition of the diets are now included in the methods section.
2.1.1. Materials
Chow diet (Irradiated Rat and Mouse Diet) and HFD (SF04-001) were obtained from Specialty Feeds (Western Australia). Both diets were produced as cylindrical pellets with a diameter of 12mm and comparable fiber contents of 5.2% and 5.4% respectively. The standard chow provides 12% of digestible energy from fat, 23% from protein and 65% from carbohydrates and contained 0.78% saturated fats, 2.06% monounsaturated fats and 1.88% polyunsaturated fats by weight. The HFD provides 43% of calories from fat, 21% from protein and 36% from carbohydrates, and contained 10.03% saturated fats, 8.24% monounsaturated fats and 5.11% polyunsaturated fats by weight. Both diets were wheat- and soy-based, but differed in the primary source of fat; namely, fish meal, mixed vegetable oils and canola oil for the standard chow or lard and soybean oil for the HFD. Deoxycholate was obtained from Sigma (Missouri, USA).
Furthermore, we added a sentence comparing the diets in section 3.1:
Chow and HFD diets had comparable fiber (5.2% vs 5.4%) and protein (23% vs 21%) content, but the digestible energy from fat increased from 12% in chow to 43% in HFD, while carbohydrate reduced from 65% to 36%.
Round 2
Reviewer 1 Report
The authors have explained all my questions and I agree that this current manuscript can be published in Biomolecules.